# Characterization of Carbon-Black-Based Nanocomposite Mixtures of Varying Dispersion for Improving Stochastic Model Fidelity

**DOI:** 10.3390/nano13050916

**Published:** 2023-03-01

**Authors:** Tyler Albright, Jared Hobeck

**Affiliations:** Alan Levin Department of Mechanical and Nuclear Engineering, Kansas State University, Manhattan, KS 66506, USA

**Keywords:** stochastic modeling, carbon black, microstructure, piezoresistivity, thin-films

## Abstract

Carbon black nanocomposites are complex systems that show potential for engineering applications. Understanding the influence of preparation methods on the engineering properties of these materials is critical for widespread deployment. In this study, the fidelity of a stochastic fractal aggregate placement algorithm is explored. A high-speed spin-coater is deployed for the creation of nanocomposite thin films of varying dispersion characteristics, which are imaged via light microscopy. Statistical analysis is performed and compared to 2D image statistics of stochastically generated RVEs with comparable volumetric properties. Correlations between simulation variables and image statistics are examined. Future and current works are discussed.

## 1. Introduction

The development of carbon-black-based (CB) composites dates to the 1890s, when Binney and Smith combined their patented amorphous CB with various waxes to create the first paper-wrapped black crayon marker [1]. The usage of CB as a composite filler material expanded to products such as automotive tires and high-voltage wire coverings, and it is still commonly used today. The effect of dispersion on the mechanical and electrical properties of these newly discovered composite materials was quickly realized, and investigations into CB dispersion in suspending matrices can be found as early as 1931 [2]. Boonstra and Medalia described the process of mixing as “intangible” and “of the most variable” in CB-rubber technology [3]. Research regarding the fundamental problem of CB-polymer dispersion persisted through the 20th century. For an in-depth review of the field during this time, see the work of Huang [4].

Since the turn of the century, research interest regarding the unique electromechanical properties of carbon-based nanocomposites, as well as their potential engineering applications, has grown rapidly. Researchers have demonstrated the applicability of CB-based nanocomposite sensors for chemical vapor detection, strain sensing, and structural health monitoring [5,6,7]. Dispersion influence on the properties of nanocomposites is frequently noted in the recent literature, especially in experimental studies observing electromechanical behaviors of carbon-based nanocomposites [8,9,10,11,12,13,14,15,16]. Some studies examine dispersion influence trends in experimental data by comparing properties of equivalently concentrated samples that differ in preparation [9,14,15]. Ji et al. defined a particle dispersion index for CB nanocomposites by curve fitting linear functions to normalized storage and loss moduli data [12]. Direct measurement and quantification of dispersion in nanocomposites has been an active area of research over the past few decades. Researchers have leveraged various technologies in the characterization of nanocomposite dispersion including small angle neutron/X-ray scattering [17], electron microscopy [8,9,17,18,19,20,21,22,23,24,25,26,27,28], atomic force microscopy [23,29], and light microscopy [24,30,31,32,33]. Microscopy images of nanocomposite samples are often used to quantify dispersion-related statistics such as the area fraction of agglomerates in the image [33], the distribution of the areas of the agglomerates in the image [19,20], estimations of average effective agglomerate diameter/shape (sphericity, convexity, and so on) [17,18,26], and other dispersion-related calculations [27,30,31]. Sample preparation methods are dependent on the interrogation tool deployed. For example, freeze-fracture and microtome sectioning is typically used to perform scanning or transmission electron microscopy on cured nanocomposite samples. Spin coating has also been demonstrated as a cost-effective option for producing nanocomposite films with optical properties favorable for dispersion interrogation via light microscopy [21,25,26,32,34,35,36,37,38].

It has been proposed that computational modeling methods might allow for more rapid investigation of the effects of dispersion on the electromechanical properties of various CB nanocomposite configurations at different length scales [39]. Feng et al. demonstrated the influence of dispersion on nanocomposite conductivity using the large-scale atomic/molecular massively parallel simulator developed by Sandia National Laboratories [40]. Akram et al. constructed COMSOL models of multilayer silicon dioxide nanocomposites from high-resolution electron microscopy images and compared the multi-physics simulation predictions of electrical permittivity to the measured permittivity of physical samples [41]. Coupette et al. developed a Monte Carlo simulation tool to generate fractal CB agglomerates in a representative volume element (RVE), and subsequently examined probabilistic network formation and percolation within it [42]. Asylbekov et al. utilized discrete element and computational fluid dynamics methods to study the fracture and dispersion of fractal CB aggregates, of varied fractal dimension and size, subjected to shear stresses comparable to those created via a planetary mixer (3 to 40-kPa) [43]. The study concluded that, with increasing shear stress, the diameter of gyration of the simulated aggregates converged to a value of 200 nm, no matter the initial size and fractal dimension defined at the start of the simulation.

Furthering the fundamental understanding of structure–property relationships of nanocomposites at length scales ranging from nano to macro will enable rapid improvement and deployment of nanocomposite sensor designs [44]. The efficiency of computational methods for testing a multitude of nanocomposite configurations and features at varying length scales is promising. Stochastic modeling methods provide a means to validate theoretical models of percolation, tunneling conductivity, and piezoresistivity with the precondition that the RVEs generated are of sufficient fidelity. In a previous article, the authors reviewed the current state-of-the-art in nanocomposite stochastic modeling and demonstrated a lack of efficiency and fidelity of a custom particle-by-particle RVE generation algorithm [39]. In this study, the fidelity and efficiency of a fractal aggregate stochastic modeling placement algorithm for spherical element nanocomposites is investigated. A high-speed spin-coater is deployed for the creation of nanocomposite thin films of varying dispersion characteristics. CB-based nanocomposite films are imaged via light microscopy, and statistical analysis is performed. The resulting 2D image statistics are compared to image statistics from stochastically generated RVEs with comparable volumetric properties. Correlations between simulation variables and image statistics are examined. Future and current works are discussed in closing.

## 2. Materials and Methods

### 2.1. Nanocomposite Spin-Coated Films

The nanocomposite mixture examined in this study is composed of a specially conductive CB (Vulcan XC Max 22, Cabot Corporation) and a two-part room cure epoxy (EZ Lam 60-min, ACP Composites). These are the same materials used by the authors in previous related works [39,45]. A master batch of CB and resin was made by first measuring quantities using an analytical balance and then combining constituents by hand-stirring before final mixing via a three-roll mill (Torrey Hills). The content of CB in the master batch was controlled to produce cured nanocomposite films containing approximately 0.25% CB by weight. Quantities of the master batch were isolated in a beaker and a thinning agent (200-proof ethanol) was added at a ratio of 0.5 parts thinner to 1 part resin. The sample was further agitated in a precise, varied, and quantifiable manner via an ultrasonic homogenization probe (Boshi Electronic Instrument) prior to the addition of the hardening agent. The hardening agent was then added to the mixture, and the mixture was hand-stirred prior to being deposited on a translucent substrate via a custom spin-coater.

A spin-coater consisting of a brushless DC motor, an electronic speed controller, and 3D printed supports was constructed as a part of this study. Precision quartz cover slips (No. 1.5H), cleaned with 98% isopropyl alcohol and mounted on laboratory microscope slides (AmScope) using transparent super glue (Loctite), served as the substrate upon which the nanocomposite films were spun. The adhered precision cover slip substrate layer was found to be necessary because the microscopy slides used in this study had surface roughness features of the order of the thickness of the spin-coated CPC films. Kapton tape covered regions of the glass slide and cover slip substrate maintained for removal of slope bias from the resulting film thickness measurements obtained via a stylus profilometer (Dektak 150). Pristine-mounted samples were first topographically mapped using the profilometer. The surface pristine samples were then plasma cleaned (Harrick Plasma) to improve the wettability of the substrate. As shown in Figure 1 below, initial spin-coating attempts with the aforementioned composite constituents produced non-uniform films susceptible to droplet formation and edge pooling. To combat this phenomenon, a 3D printing grade photopolymer resin (Anycubic) was incorporated into the mixture by hand stirring at a ratio of 1 part phenolic resin to 1 part photopolymer coincident with the addition of the hardening agent just before the deposition of the films. Immediately following deposition, the samples were exposed to high-intensity UV light (λ = 405 nm, 40 W) in order to reduce the mobility of the film during cure. After the films cured for 24 h, the samples were topographically mapped again. Comparing before and after topography maps allowed the thicknesses of the cured film to be quantified. The resulting films were then imaged via a light microscope (AmScope ME520T), equipped with an autofocus camera (Imaging Source—DFK MKU130-10x22), at precise locations navigated to by a custom-built motorized microscopy stage. The resulting images were post-processed using the ImageJ particle analysis algorithm [46].

### 2.2. Stochastic Modeling of Fractal CB Nanocomposite RVEs

The fractal nature of CB aggregates is well documented, and a comprehensive literature review of this topic is presented by Donnet et al. [47]. In previous works [39,45], the authors characterized the morphology of the CB used in this study via TEM. Aggregates of spherical particles were observed with similar colloidal structures to those observed in the text by Donnet et al. [47]. In keeping with that work, the term aggregate (Figure 2b) will be used to refer to a collection of CB spherical primary particles (Figure 2a). The term agglomerate will be used to refer to collections of aggregates that appear to be contacting. Research on the computational modeling of fractal aggregates is vast, and a comprehensive review of the field is offered by Meakin [48]. In the current study, a fractal aggregate generation tool, FracVal [49], was used to generate 10,000+ aggregates of monodisperse spherical particles whose coordinates were stored for placement in a finite and discretized RVE via adapted stochastic modeling software developed by the authors. The number of primary particles per aggregate, particle radius, fractal dimension, and pre-factor parameters used to generate these aggregate coordinate files were 100, 15 nm, 2.7, and 0.7, respectively. These values fall within ranges reported in the recent literature and were held constant for purposes of exploring other pertinent simulation parameters [42].

Evaluating the conductivity of simulated CB nanocomposite RVEs requires careful determination of the relative separation between neighboring primary particles for defining electrically conductive pathways and their resistances. The resistance of electrically conductive pathways consists of intrinsic resistance between fused primary particles in aggregates and tunneling resistance between non-fused primary particles separated by a small distance by a dielectric material (i.e., the polymer matrix) [50]. Simmons defined an exponential relationship of tunneling resistance between two conductors as a function of their separation distance and the electrical properties of the material between them [51]. Sun et al. defined a range of effective tunneling distance thresholds for various polymers [52]. These distances, ranging from 2 to 3 nanometers, constitute a threshold of separation of primary particles beyond which conductivity is effectively zero considering Simmons’ generalized formulae. In this study, the effective tunneling distance between simulated CB primary particles is defined as 3 nanometers, which is used as a threshold for determining when aggregate pairs are considered agglomerated.

The placement of primary particles while successfully documenting neighboring particles considered to be electrically conducting has been accomplished in a variety of ways, and is explained in great detail in the authors’ previous works [39,53]. One simple method is to pick a random coordinate in the RVE for placement of a new particle and calculate the distance between that particle and all previously existing particles in the RVE. This pick-and-check method requires looping through each particle to determine whether the placement of a particle at that index is valid or invalid. Invalid placement occurs when a particle is placed too close to a previously placed particle, resulting in unrealistic primary particle penetration. The authors previously showed that the computational time cost of this method increases exponentially with the increasing number of particles placed [39]. Instead of parallelizing this procedure, the authors developed a rapid and memory-intensive particle placement algorithm. Consider the cubic RVE illustrated in Figure 2c. Each discrete location in the RVE is assigned a unique numbered index. Bit-fields are then created to store binary information about each index, including the invalidity of placing a particle at that index (visually represented by red indices in Figure 2a), the existence of a particle centered at that index, and whether or not a particle placed at that index would be in close proximity to pre-existing particles in the RVE (visually represented by green indices in Figure 2a,b). A variable length array containing indices of valid and neighboring placement locations is constantly updated via a simple relative index-mapping scheme once particles are added to the RVE. This method allows for rapid identification of valid, invalid, and agglomerate producing placement locations without recalculating interparticle distances every time a new particle is placed.

As noted by Coupette et al., it is difficult to experimentally distinguish agglomerates from aggregates in bulk materials [42]. Van der Waals attraction between aggregates results in weakly linked agglomerates, varying in apparent size and shape. To recreate this phenomenon, the agglomeration of aggregates must be controlled via simulation parameters. Relevant simulation parameters include the weight fraction of CB to be placed in the RVE, the physical properties of the CB and polymer, the dimensions of the RVE, the unit length of discretization, and the number of aggregates to be placed at random and via forced agglomeration. After defining the simulation parameters, each coordinate file from the bank of FracVal aggregates is loaded into memory. Each particle in the FracVal aggregate file is assigned a relative index for rapid integration into the global coordinate/indexing system of the RVE. The placement algorithm begins by randomly placing a user-defined number of seed aggregates in the RVE. Once the seed aggregates have been placed, a percentage of the remaining particles are placed as FracVal aggregates using a forced agglomeration sub-routine. In this sub-routine, a random index in the volume is selected, followed by the selection of a random aggregate from the collection of FracVal aggregates. The relative aggregate indices are transformed according to the selected random index, and a loop is used to determine if (1) all of the placement indices of the new aggregate are valid, and (2) if any of the particles have been placed in close proximity to particles of previously placed aggregates. If both (1) and (2) are true, the placement is considered valid, the bit-fields are updated, and the next aggregate is placed. Following the placement of the aggregates in forced agglomeration, individual particles are placed in the RVE under forced agglomeration until the desired weight fraction is met. These individual particles constitute the fill percentage variable *f*, which is defined as
(1)$$f=\left(1-\frac{N*100\left[\frac{particles}{aggregate}\right]}{T-S*100\left[\frac{particles}{aggregate}\right]}\right)*100\%$$
where *T* is the total number of particles in the RVE, *S* is the total number of seed aggregates, and *N* is the total number of aggregates placed via forced agglomeration. After all particles have been placed, the coordinates of each particle are saved for post-processing. Ovito is then used to render the RVEs and generate image files [54]. The resulting image files are statistically analyzed using ImageJ, and image statistics are compared to those observed via experimentation [46].

## 3. Results

### 3.1. Influence of CPC Film Thickness on Observed Agglomeration

The resulting thickness of spin-coated polymer solutions is affected by the rotational speed of the sample, physical properties of the constituents in the solution, and ambient conditions, among other factors [55]. In this study, the concentrations of CB, resin, hardener, and ethanol were kept constant in all experimental samples, while the effects of variations in the preparation and deposition of the spun mixtures were investigated. First, the effect of rotational speed on the thickness and optical characteristics of the nanocomposite films was investigated. Figure 3 shows microscopy images of spin-coated nanocomposite samples dynamically deposited at varying rotational speeds and spun for 60 s at a constant speed after deposition. CB-resin from the master batch was diluted with ethanol and the mixture was homogenized via an ultrasonic probe for 12 min with an estimated applied energy of 17 kJ (considered very well mixed), a quantification/characterization that will be explored in greater detail later in this paper. The rotational speed of the spin-coater stage was measured using a strobe tachometer. The average thickness of five cured film samples was used to determine the average film thickness for each plotted point in Figure 4c. From the microscopy images in Figure 3a–c, there are clear differences in the apparent size and distribution of CB agglomerates between films of varying thickness. As film thickness increases, the agglomerates in the images appear to grow in size. Matching stochastic film RVEs were generated for comparison (Figure 3d–f), and the trend was replicated. The 6 μm thick stochastic film RVE (Figure 3d) contained approximately 9 million CB particles, consumed 115 GB of memory, and took just less than 3 h to complete. For comparison, the 2 μm thick stochastic film RVE (Figure 3f) contained 3 million CB particles, consumed 35 GB of memory, and took just less than 1 h to complete. While the authors acknowledge the influence of thickness on the optical characterization of dispersion of CB in bulk nanocomposites, computational demand was influential in the decision to limit the scope of this study to nanocomposite films less than 2 μm in thickness.

### 3.2. Validation of the Thin Film Topography Characterization Method

A topographical map of a cured nanocomposite film sample presented in Figure 4a shows an example of the before and after spin coating scans of a very well mixed sample, and highlights the necessity for slope correction. This difference in slope is a result of the waviness of the profilometer stage surface and the required removal, handling, and replacement of each sample on the stage in between scans. To quantify the inherent error of this measurement process, a neat sample was scanned 10 times while being removed and replaced at various locations on the profilometer stage. Using the stationary camera on the profilometer, the machined edge of the glass microscope slide was aligned with the profilometer XY-axes and the machined corner of the sample was used as a reference location for the start of each scan. All scan data were collected in Python for assessing differences in the reported datasets. The maximum and minimum reported values from each scan at each measurement location (after slope correction) were recorded, and the greatest difference between those pairs was found to be approximately 50 nm. Each of the 10 individual measurement profiles was compared to the average of all 10 profiles, which yielded an average difference of approximately 30 nm. Considering that the average thicknesses of the films in the dispersion characterization study ranged from 1.5 to 1.8 μm, the average thickness measurement accuracy was found to be within ~2%. As is apparent in Figure 4b, the thickness of the deposited films also varies along the cover slip. This thickness variation is most notable at the peaks and valleys of convex and concave regions of the substrates. In an attempt to quantify the variation in film thicknesses caused by substrate waviness, 10 samples were spin-coated at a constant speed of 10,000 rpm. The mixture deposited was once again considered very well mixed. Of the 10 samples spun, the average standard deviation of the thickness was approximately 350 nm. Thickness measurements within 3 mm of the cover slip edges were omitted from this variance estimation, as edge buildup can be observed in the plots of Figure 4. These edge regions were also excluded during microscopy imaging. Excluding edge regions, the film samples were divided into smaller target locations where imaging was favorable. The coordinates of each target location (relative to the sample datum) were stored in a file along with the average thickness and other relevant statistics that were later accessed by a script controlling the motorized XY stage for rapid navigation to favorable imaging regions. The total area captured via microscopy at each location depends on the magnification used, and determines how many images each sample can produce without oversampling. Magnifications of 10×, 40×, and 100× capture circular images of approximately 4 mm, 1 mm, and 0.5 mm in diameter, respectively. Square images are then cropped from the center of those circular images (i.e., 10× cropped to 2.5 mm side length) for post-processing and statistical analysis. 

### 3.3. Statistical Analysis of Agglomeration in CPC Thin Films of Varied Mixture Quality

Dispersion energy applied via ultrasonic homogenization was varied between 0 kJ and 25 kJ. Five film samples were generated at each dispersion increment while maintaining mixture constituent concentrations. Figure 5 shows microscopy images from two separately prepared mixtures with applied dispersion energies of 2 kJ (Figure 5a,b,d) and 17 kJ (Figure 5b,e,f), acquired at two magnifications. Comparing Figure 5a,b, one can confidently make a qualitative characterization of the differing dispersion states—that is, Figure 5a is considered poorly dispersed and Figure 5b is considered very well dispersed. Closer inspection of the same samples at a greater magnification (i.e., Figure 5c,e) reveals dispersion states that are more difficult to distinguish. A moderately large agglomerate feature in Figure 5a is magnified in Figure 5d. Numerical determination of the exact boundaries of such agglomerates captured at 100× magnification proved difficult. Image contrast algorithms were deployed to improve the particle analysis macro’s ability to capture all visible agglomerates. The success of those algorithms was determined by manual comparison of visible agglomerates counted in grayscale microscopy images (Figure 5e) to their binary transformations (Figure 5f). Often, those same contrasting algorithms were unsuccessful when deployed on images of poorly dispersed regions (Figure 5d). In such cases, visible agglomerates were omitted or boundary noise was generated as a result of inconsistent edge contrasting, a product of imperfectly leveled samples and shadowing/blurring during imaging. For this reason, agglomerates found to be contacting image boundaries were excluded from the statistical analysis.

Statistics from images acquired at 100× magnification revealed a similar particle area and perimeter distributions for samples across the entire range of dispersion energies. The vast majority of recorded agglomerate areas from those images ranged from 1 to 50 μm^2^. At 10× magnification, agglomerates of a similar order of magnitude in area were difficult to distinguish from background noise, as shown in Figure 6b—a close-up image of a well-defined agglomerate from Figure 6a. Surrounding this agglomerate, one can identify collections of slightly darkened pixels, ranging from areas of 1 to 4 pixels^2^, or 13.5 to 216 μm^2^ considering the scale of the image. With knowledge of the range of agglomerate areas defined at a higher magnification, one could reasonably assume that these dark collections of pixels are in fact CB agglomerates. Programmatic boundary definition for agglomerates of such size introduces a greater probability of noise (owing to image digitization); therefore, agglomerates with areas under a minimum area threshold were filtered from the datasets. Statistical analysis was performed while varying the applied minimum area threshold to observe the effects of such data filtering, and those results are shown in Figure 6c,d. Approximately 250 images were acquired from each sample at 100× magnification, which amounts to a cumulative imaged area of approximately 20 μm^2^ per sample. At 10× magnification, 40 images were acquired from each sample, resulting in a cumulative imaged area of approximately 250 μm^2^ per sample. By imaging a larger area at a lower magnification, the largest agglomerates on each sample are typically captured, thus greatly influencing the averages reported in Figure 6c. Consider an image of a poorly mixed sample (Figure 5a) containing large agglomerates. When focusing in on larger agglomerates at 100×, often, agglomerates are so large that the entire image is black. Such images, which would drastically skew the averages of their datasets, are typically omitted through the exclusion of edge agglomerates. The average agglomerate areas, from images acquired at 10× and 100× and with various threshold filters applied, are plotted in Figure 6c and Figure 6d, respectively. From Figure 6d, no consistent trend between the average agglomerate area and sample probe dispersion energy is observed. Images at 10× magnification revealed drastic differences in dispersion between samples prepared with varied energies applied via the ultrasonic probe. In Figure 6c, the data suggest that the average agglomerate area tends to decrease with increased mixture dispersion energy. Samples prepared with 5 kJ or greater dispersion energy showed consistent average agglomerate areas when applying a constant minimum area threshold. Average agglomerate areas of the datasets filtered with the smallest minimum area threshold, 4 pixels^2^ or 55 μm^2^, were found to be consistent across all samples. As the minimum area threshold increased, the average agglomerate area of samples prepared with 5 kJ or greater dispersion energy tended to scale similarly, while the average areas of less dispersed samples tended to increase much more rapidly. This trend is attributed to the fact that the majority of CB agglomerates have apparent areas that cannot be discerned from background noise at 10× magnification. It is concluded that data from microscopy images acquired at 10× magnification more precisely define the positive tail end of the agglomerate area distribution curve at the macro scale. This tail end of the distribution is where differences in dispersion characteristics are apparent to the naked eye and easily quantified.

### 3.4. Exploration of Stochastic Model Agglomeration Tunability

Simulating RVEs with 2D dispersion characteristics comparable to those seen in Figure 5a presents a unique challenge that the current stochastic simulation tool is not suited to efficiently address. In such poorly dispersed samples, the concentration of CB in large agglomerates greatly skews the relative densities and weight fractions of CB at various locations about the film. Such large agglomerates cannot be replicated using the current simulation configuration; however, the current simulation configuration can be used to accurately represent well-dispersed CB nanocomposite films, such as those shown in Figure 3. Thousands of RVEs, of comparable thicknesses to experimental film samples, were generated while varying simulation placement algorithm parameters. Statistical analysis was performed on the resulting RVE cross-sectional images and compared to statistics from experimental film images in Figure 7. 

The imaged experimental film samples were prepared with ultrasonic dispersion energies greater than 15 kJ and imaged at 100× magnification. Distribution plots of agglomerate area, perimeter, circularity, roundness, solidity, and aspect ratio (AR) for experimental film samples (solid lines) and simulated film RVEs (dashed/dotted lines) are compared in each plot in Figure 7. These image statistics are directly reported from the ImageJ particle analysis macro, and technical formulations can be located via official online ImageJ documentation [46]. Figure 7 demonstrates the capability of the model to generate a large range of agglomerated fractal aggregate structures that are comparable to cross-sectional images of experimental film samples. Variations in the geometries of such agglomerates can be quantified using commonly deployed image analysis methods, and user-defined model parameters can be tuned to replicate geometries and shape descriptor statistics observed in experimental samples. Advanced numerical techniques, such as genetic algorithm tuning [56] and information entropy analysis [57,58], could also be deployed in the proposed schema to more precisely recreate these complex particle systems. The number of possible placement parameter and FracVal aggregate generation parameter combinations is vast, and precise tuning of these simulation variables is beyond the scope of this paper. Determination of the correct combination of variables to systematically replicate experimentally derived image statistics poses a challenging multivariate non-linear regression problem, which will be the focus of future work. 

## 4. Conclusions

This research is motivated by a long-term goal to further our understanding of the influence of particle dispersion on electromechanical properties of nanocomposites. Important steps toward achieving this goal—and the main contributions of this paper—are as follows: (1) creating an accessible framework for quantifying dispersion quality and (2) efficiently generating high-fidelity RVEs that are statistically comparable to real mixtures. In summary, a rapid, scalable, memory-intensive stochastic modeling placement algorithm was developed for generating fractal CB agglomerate-based nanocomposite RVEs. A method for producing CB-based nanocomposite films and quantifying dispersion of the mixtures was demonstrated. Experimental samples of varying dispersion were imaged at magnifications ranging from 10× to 100×. Lower magnification imaging revealed significant differences in CB agglomerate size distribution, specifically differences at the positive tail end of the distribution. Microscopy images acquired at a higher magnification revealed very similar statistical distributions of agglomerate area for samples of dissimilar dispersion characteristics. Such similarities are partially attributed to down sampling and the varying magnification scales used in constructing image statistics to cover the range of agglomerate sizes. Experimental sample film thicknesses were characterized via stylus profilometry, and RVEs with similar volumetric properties were generated for comparison. Statistical analyses of 2D images of stochastically generated film RVEs were directly compared to experimental samples. The correlation between simulated agglomerate shape descriptors and stochastic placement algorithm parameters was demonstrated.

Current and future research is focused on model parameter tuning via artificial neural networks and other modern machine learning concepts. Measurable properties, such as electrical and thermal conductivity, are promising candidates for tuning stochastic simulation parameters. Electrical conductivity and piezoresistivity models for stochastically generated film RVEs are currently under investigation.

## Figures and Tables

**Figure 1 nanomaterials-13-00916-f001:**
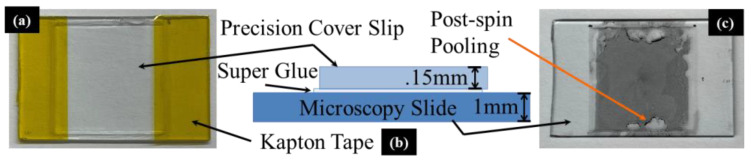
(**a**) Clean sample prepared for spin coating; (**b**) cross-sectional view of sample layer configuration; (**c**) sample post deposition of nanocomposite mixture exhibiting undesirable pooling.

**Figure 2 nanomaterials-13-00916-f002:**
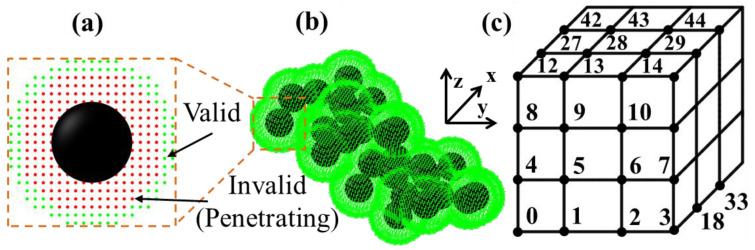
Simulated 3D CB structures of (**a**) an aggregate of fused carbon particles, (**b**) a primary particle with mapped valid (green) and invalid (red) placement indices, and (**c**) illustrated RVE with numbered indices.

**Figure 3 nanomaterials-13-00916-f003:**
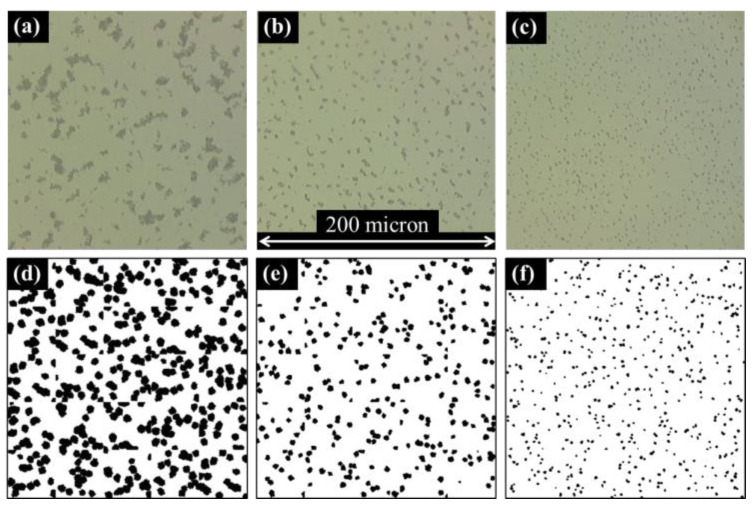
Microscopy images at 100× magnification of spin-coated films with average thicknesses of (**a**) 6 μm, (**b**) 4 μm, and (**c**) 2 μm; (**d**–**f**) simulated film RVEs of matching thickness and cross-sectional area to their experimental counterparts above.

**Figure 4 nanomaterials-13-00916-f004:**
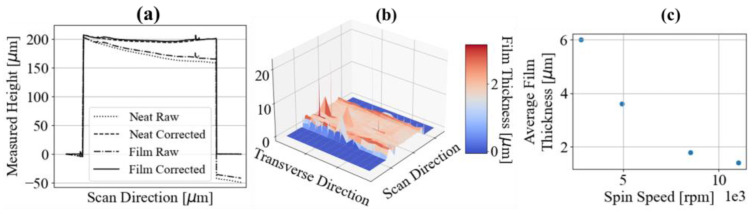
(**a**) Profilometry scans at the same location of the neat and post-film deposition sample. Raw scan data are displayed to demonstrate the importance of slope correction; (**b**) surface plot of stitched profilometry scans; (**c**) average film thickness versus rotational velocity of the sample during spin coating deposition and spinout.

**Figure 5 nanomaterials-13-00916-f005:**
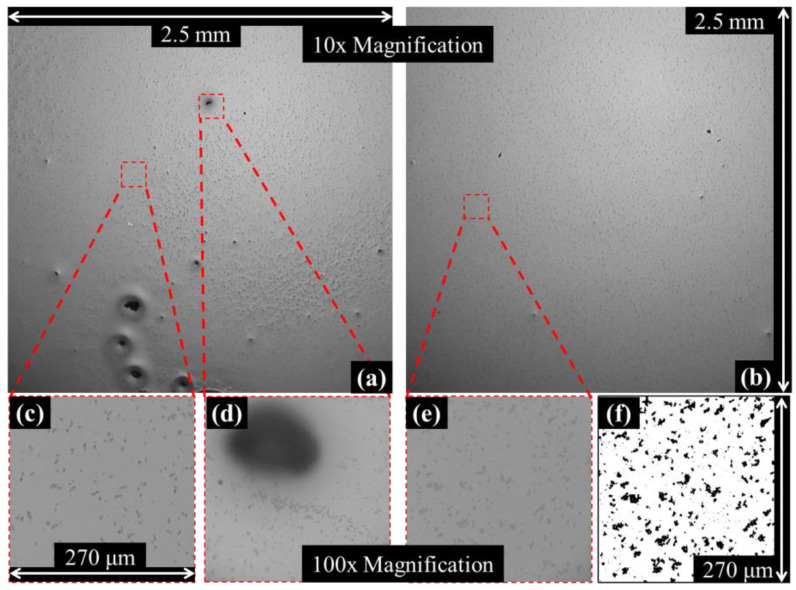
Microscopy of two separately prepared nanocomposite film samples at varying magnifications. (**a**,**c**,**d**) Sample probed with 2 kJ of energy, imaged at (**a**) 10× magnification and (**c**,**d**) 100× magnification. (**b**,**e**,**f**) Sample probed with 17 kJ of energy, imaged at (**b**) 10× magnification and (**e**,**f**) 100× magnification.

**Figure 6 nanomaterials-13-00916-f006:**
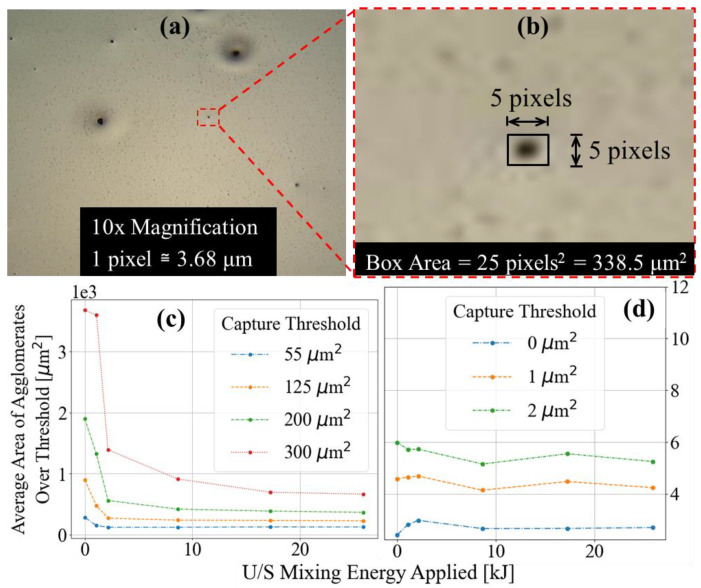
(**a**) Moderately mixed (2.16 kJ) nanocomposite film sample imaged at 10× magnification and (**b**) enlarged to show pixilation of agglomerate features and their approximate dimensions. Plots of average agglomerate areas of samples with varied dispersion energy at (**c**) 10× and (**d**) 100× magnification.

**Figure 7 nanomaterials-13-00916-f007:**
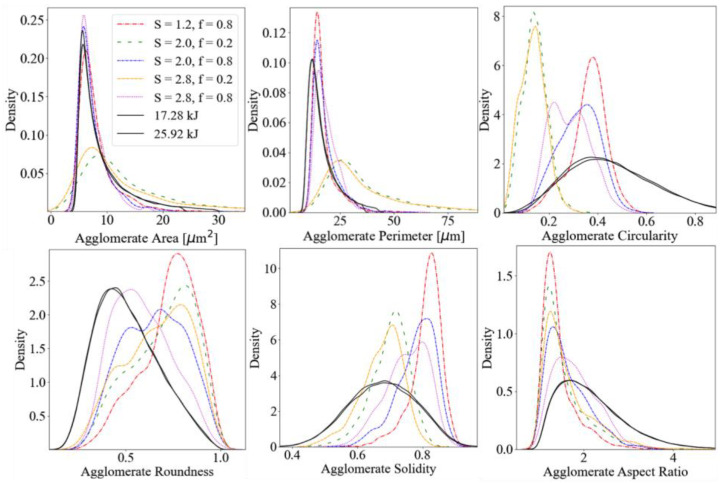
Kernel density estimate plots of various shape descriptor statistics from ImageJ particle analysis of collections of microscopy images of well dispersed samples imaged at 100× magnification (solid lines) and collections of simulated film RVEs (dashed/dotted lines) with varying stochastic model agglomerate placement parameters.

## Data Availability

The source code developed and deployed to generate the data presented in this paper has been made publicly available in the following GitHub repository: https://github.com/tylerbalbright/Stochastic_Model_CB_FracVal (accessed on 10 February 2023).

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
