# Peer review of "Characterization of Carbon-Black-Based Nanocomposite Mixtures of Varying Dispersion for Improving Stochastic Model Fidelity"

_nanomaterials, 2023, doi:10.3390/nano13050916_

Round 1

Reviewer 1 Report

The manuscript deals with developing the numerical model for assessing the dispersion of the carbon-black agglomerates. The latter is necessary for appropriate treating experimental results and obtaining high-reproducibility carbon materials. I think the manuscript is quite interesting, intending to solve an important task of carbon materials science, and closely relating to experiments. The manuscript is well organized and well supported with auxiliary materials. However, I recommend it after some improvements.

The questions to improve the quality of the manuscript are following.

(1) The authors freely operate with the concept of fractals. In general, not every branched structure is fractal. There are mathematically strict definitions of the fractals. As for materials science, this is the condition of the similarity between the structural patterns of the different hierarchical levels of the structure organization. How the authors confirm and quantify the fractal structure of the experimental samples of the carbons? Or fractals appear only in the mathematical model. Please, clarify.

(2) The author should mention other approaches to characterizing fractal nanoparticles, for example, very recent approach based on the information entropy applied to the particle analysis a priori or a posteriori. See, respectively, Sabirov et al. Information Entropy of Regular Dendrimer Aggregates and Irregular Intermediate Structures. Liquids 20211(1), 25-35, https://www.mdpi.com/2673-8015/1/1/2; McFhionnlaoich & Guldin. Information Entropy as a Reliable Measure of Nanoparticle Dispersity. Chem. Mater. 2020, 32 (9), 3701–3706, https://pubs.acs.org/doi/full/10.1021/acs.chemmater.0c00539).

(3) As follows from the figures, the agglomeration results in compact structures. In this regard, is it possible to apply here one old numerical parameter used in geology, that is sphericity (Ψ). It is defined as Ψ = π1/3(6V)2/3/A, where V and A are the volume and the surface area of the particle. Sphericity is the ratio of the surface area of a sphere, having the same volume as the given particle, to the surface area of the particle. See: Wadell, H. (1935) Volume, shape and roundness of quartz particles. J. Geol., 43: 250–280; https://www.journals.uchicago.edu/doi/10.1086/624298.

Reviewer 2 Report

In this study, nanocomposite films with different dispersion properties were imaged by light microscopy and a stochastic fractal aggregate placement algorithm was used for statistical analysis. The correlation between simulated variables and image statistics was investigated. Finally, the methodology and concerns of this paper are appealing and the authors were asked to further answer the questions I posed and revise the manuscript. If the authors are willing to revise the manuscript, I would like to recommend the publication of this manuscript.

1.      In the introduction section, the authors mainly state the means of characterizing the structure-property relationship of nanocomposites. However, the focus of this study is on the stochastic modeling placement algorithm, and the innovative points of this algorithm are not highlighted. Also, the limitations regarding the previous statistical methods are not described.

2.      When describing the subroutine loop in 2.1, the author describes the judgment condition and indicates that the placement is considered valid if the judgment result is true. It would be helpful to provide a program logic diagram for understanding the program.

3.      Line 234, I could not find figure 3G in the manuscript.

4.      Figure 3 is followed by Figure 1, the picture numbering is not correct.

5.      “For this reason, agglomerates found to be contacting image boundaries were excluded from statistical analysis.” I am curious about how the authors ensured that the extent of the excluded boundary clumps was justified without enlarging the area of the shaded region.

6.      “Distribution plots of agglomerate area, perimeter, circularity, roundness, solidity, and aspect ratio (AR) for experimental film samples imaged at 100x magnification (solid lines) and comparable simulated film RVEs (dashed/dotted lines) are shown in Figure 7.” However, the authors do not describe these images and the reader is not able to understand what information the corresponding curves show or what features they reflect. It also does not further indicate the level of sophistication of this algorithm, and the authors were asked to further indicate the need for this figure.

7.      I would like to know if the algorithm in this paper is portable. Can it be used for other models or light mirror images, or is it only applicable to the model in this paper. And this point needs to be mentioned in the manuscript.

Round 2

Reviewer 1 Report

The authors have appropriately revised the manuscript. Hence, it is recommended for publication in Nanomaterials.